# Attitudes, Perceptions and Practices of Influenza Vaccination in the Adult Population: Results of a Cross-Sectional Survey in Spain

**DOI:** 10.3390/ijerph191711139

**Published:** 2022-09-05

**Authors:** Camino Prada-García, Virginia Fernández-Espinilla, Cristina Hernán-García, Iván Sanz-Muñoz, José Martínez-Olmos, Jose M. Eiros, Javier Castrodeza-Sanz

**Affiliations:** 1Department of Preventive Medicine and Public Health, University of Valladolid, 47005 Valladolid, Spain; 2Dermatology Service, Complejo Asistencial Universitario de León, 24008 León, Spain; 3National Influenza Centre, Edificio Rondilla, Hospital Clínico Universitario de Valladolid, 47009 Valladolid, Spain; 4Preventive Medicine and Public Health Service, Hospital Clínico Universitario de Valladolid, 47003 Valladolid, Spain; 5Andalusian School of Public Health, 18011 Granada, Spain; 6Microbiology Service, Hospital Universitario Río Hortega, 47012 Valladolid, Spain

**Keywords:** influenza vaccine, vaccination coverage, elderly, COVID-19, survey, Spain

## Abstract

In Spain, the 2021/22 influenza season overlapped with the sixth wave of the 2019 coronavirus disease pandemic (COVID-19). Influenza is a major public health problem associated with high morbidity and mortality. The objectives of this study were to determine the knowledge, perceptions and practices of influenza vaccination in the Spanish population, coinciding with the COVID-19 pandemic, with special attention paid to people over 65 years of age. A cross-sectional study was carried out by conducting 2211 telephone interviews. It was observed that 81.6% of people ≥ 65 years were vaccinated annually or with some frequency compared to 35.5% of those under 65 years (*p* < 0.001). Fifty percent of Spaniards showed an intention to be vaccinated in the 2021/22 campaign, during the SARS-CoV2 pandemic. In the case of people ≥ 65 years old, this figure was 83% compared to 42% of those under 65 years old (*p* < 0.001). Significant predictors of intention to be vaccinated were age of 65 years or older (OR 1.8, 95% CI 1.3–2.5), female sex (OR 1.9, 95% CI 1.5–2.4), belonging to risk groups (OR 2.2, 95% CI 1.6–3.1) and having been previously vaccinated (OR 29.7, 95% CI 22.5–39.2). The main reasons for deciding to be vaccinated were the need to be protected against the virus and to be vaccinated annually. On the other hand, lack of recommendation and considering the influenza vaccine as not necessary were the main reasons for not getting vaccinated. In addition, health personnel stood out as the main source of information (32.9%) compared to traditional media (26.9%) and public administration (12.3%). This study aimed to assess and analyse the factors influencing willingness to receive influenza vaccines in the COVID-19 era among Spanish adults, as well as the main information channels and strategies to encourage vaccination.

## 1. Introduction

Seasonal influenza is an acute respiratory infection caused by influenza viruses. There are three types of seasonal influenza viruses (A, B and C), with types A and B being the most common. It can cause annual epidemics and outbreaks that occur in different seasonal patterns depending on the region of the world. The duration of epidemics is about 4 months, although the peak incidence is reached in a period of 1 to 2 months [1]. The disease has important social and economic repercussions, making it a major public health problem [2,3,4,5]. It can affect between 5% and 20% of the general population and up to 50% of the institutionalised population, generating a significant number of medical consultations and loss of working hours as a consequence of the disease [3,6]. Globally, annual epidemics are estimated to cause 3–5 million cases of severe disease and 250,000–500,000 deaths per year [2,7,8,9,10,11]. Overall influenza-associated mortality has been estimated at 13.8 per 100,000 person-years in Europe [3]. It affects all age groups and is self-limiting in most of the population. However, in older people who are immunocompromised or have chronic diseases, it can present as a severe disease requiring hospitalisation [2,12,13,14]. Approximately 70–90% of influenza-related deaths and 50–70% of hospitalisations for serious complications, such as pneumonia, myocarditis or encephalitis, occur in the ≥65 age group [15]. Thus, older adults are one of the high-risk groups recommended by the World Health Organization (WHO) for annual influenza vaccination. This age group is more susceptible to short- and long-term complications, including hospitalisation, morbidity and mortality, compared to younger adults, due to the process of immunosenescence.

Annual vaccination has been shown to reduce the risk of death and complications in people aged 65 years and older, but influenza vaccination coverage rates are below the 75% target proposed by the WHO and the European Centre for Disease Prevention and Control (ECDC) [3,7]. In addition, the severity of influenza among the elderly is increased by the higher prevalence of comorbidities among this age group [16]. In Spain, the last seasonal influenza epidemic in 2021/22 coincided with the sixth wave of the pandemic caused by SARS-CoV-2. Both influenza and COVID-19 share several signs and symptoms, both being mainly respiratory pathologies, ranging from asymptomatic or mild to severe illness and death. In this context, the role of influenza vaccination during the period 2021/22 is of particular importance to reduce the burden of disease due to influenza, but also because of the coincidence with COVID-19, which could lead to increased pressure on healthcare systems [17]. Apart from clinical and care considerations, it is also important to consider the economic benefits of influenza vaccination. For the elderly, most published studies have found vaccination to be cost-effective, offering cost savings by reducing treatment costs that exceed the cost of vaccination [15,18,19].

In view of the above, vaccination against influenza in the 2021/22 season is particularly relevant, and the assessment of people’s knowledge and attitudes towards vaccination may provide relevant data that will help to increase vaccination coverage, especially in people in risk groups. Moreover, in this context of the coincidence of two viral respiratory processes, it is important to analyse whether healthcare professionals continue to be a fundamental pillar when it comes to providing information to patients and making recommendations, since their own opinion and example may be fundamental in increasing or decreasing vaccination coverage [20].

Therefore, the objectives of this study were to determine the knowledge, habits, attitudes and practices of vaccination against influenza in the general Spanish population coinciding with the COVID-19 pandemic and to identify the aspects that promote vaccination and the barriers that limit it, with special attention paid to people over 65 years of age, in order to design future strategies and initiatives to promote vaccination in this group.

## 2. Materials and Methods

A cross-sectional observational study was designed at the national level, for which an anonymous questionnaire was created to collect the opinions, attitudes and practices of Spanish citizens over 18 years of age regarding vaccination against influenza during the month of September 2021, coinciding with the sixth wave of the COVID-19 pandemic. This non-interventionist survey, based on participants’ opinions, was conducted in accordance with the General Data Protection Regulation (GDPR). Participation was voluntary, and GAD3, the company responsible for its implementation, was in charge of protecting the individualised information of the respondents.

### 2.1. Study Population and Sampling Frame

The geographical scope of this study consisted of all the autonomous communities, and the study population corresponded to all persons of legal age. It was carried out by means of an anonymous survey conducted by GAD3, a social research and communication consultancy firm [21], on behalf of CSL Seqirus, by means of a computer-assisted telephone interview known as computer-assisted telephone interviewing (CATI). A fixed sample distribution was followed in the autonomous communities of greatest interest (200 to 300 interviews in each territory) in order to be able to offer territorialised results. A total of 2205 random interviews were carried out, with a 50% landline/mobile phone distribution, between 17 and 29 September 2021 in the time slot from 9 a.m. to 9 p.m. The approximate duration of each interview was between 3 and 4 min.

### 2.2. Study Variables

A questionnaire of 19 questions was used, most of them closed, which included socio-demographic variables such as age; sex (male or female); current employment status (private sector worker, public sector worker, self-employed/entrepreneur, retired, unemployed, student, domestic worker); level of education (primary or basic, secondary, university); belonging to risk groups (people over 65 years of age, diabetics, cardiac pathologies, respiratory pathologies, immunosuppression, other chronic pathologies, health professionals, pregnant women, people living with patients at risk, people with disabilities, essential workers); the postcode of residence; vaccination status in previous years (annually, with some frequency, not); the most relevant reasons for deciding to be vaccinated (for my own protection and/or that of my environment, because I have sufficient information about the vaccine, because I have suffered the consequences of influenza before, because my doctor recommended it, because nurses recommended it to me, because of social responsibility) and not to be vaccinated (lack of confidence in the effectiveness of the vaccine, I consider that the flu is not serious, I do not have enough information about the vaccine, my doctor has not recommended it, I have a phobia of needles, I was previously vaccinated and it did not feel good, it has not been recommended/prescribed to me); the means of information through which the participant received some type of information about the vaccination campaign (public administration, health personnel, media, social networks, family or friends, by myself); the degree of encouragement to vaccinate (a lot, not much, not at all, none) through the following actions (people around me deciding to get vaccinated, an awareness-raising campaign in the media, facilitating access to Primary Healthcare Centres, creating “*vaccinodromes*” such as those of COVID-19, sending annual reminders via SMS, having more information about vaccination and its benefits) in relation to the upcoming 2021/22 vaccination campaign during the SARS-CoV-2 pandemic: intention to be vaccinated (yes, no), most relevant reasons for deciding to get vaccinated (it is necessary to protect against viruses; in case I have COVID-19, it will help me with the effects; it is necessary to get vaccinated every year for flu; I trust vaccines in general; for social responsibility; because COVID-19 has made me more aware of the importance of vaccination) and the most relevant reasons for deciding not to get vaccinated (I consider that flu is not a serious virus, it is enough with the COVID-19 vaccine, it is not necessary to get vaccinated every year, the COVID-19 vaccine is sufficient, vaccination is not necessary, I do not trust vaccines in general, it is not effective, it has not been recommended/prescribed to me); importance of vaccination compared to other years in the context of the COVID-19 pandemic (more important, equally important, less important); and the need for a specific flu vaccine in older people (yes in any case, yes but it depends on the price, it is not necessary).

### 2.3. Statistical Analysis

With the sample obtained, a descriptive statistical analysis of the data resulting from the responses to the questionnaire was carried out. The study was designed with a sampling error of ±2.1% for a degree of confidence of 95.5% (two *sigmas*) and under the worst-case hypothesis of P = Q = 0.5 under the assumption of simple random sampling. The study of the association between the variables was carried out using Pearson’s chi-square test. Multivariate logistic regression analysis was performed to identify predictors of the intention to vaccinate against influenza. Odds ratios (ORs) were calculated for the variables analysed by logistic regression with 95% confidence intervals. The level of significance was set at *p* < 0.05. The statistical software IBM SPSS Statistics version 25 was used (IBM Corp, Armonk, NY, USA).

## 3. Results

In order to describe the findings, the results are divided into three main sections: the first section details the socio-demographic variables of the study; the second section refers to the attitudes and practices of the population regarding vaccination against influenza before and during the COVID-19 pandemic; and the third section mentions the most frequently used means of information and the measures to encourage vaccination.

### 3.1. Socio-Demographic Variables of the Study

Of the 2205 people interviewed, 51.7% were women and 48.3% were men. In total, 27.0% were aged 65 and over. In terms of employment status, almost 30% were workers in the private sector, while 30.4% were retired.

In terms of belonging to risk groups, one third of Spaniards said that they belonged to one. Of these, 37% were over 65 years of age, and 17% suffered from respiratory pathologies. The sample was considered to be representative of the Spanish adult population in terms of the main socio-demographic characteristics (Table 1).

### 3.2. Attitudes and Practices Regarding Influenza Vaccination

Analysis of previous vaccination status showed that one third of Spaniards were vaccinated against influenza annually, and 52% had never been vaccinated. Up to 34.7% of Spaniards who did not belong to any risk group were vaccinated annually or with some frequency compared to 77.5% of those who did belong to a risk group (*p* < 0.001). 

When studying people aged 65 years of age and older, 69.1% were vaccinated annually, and up to 81.6% of people in this group were vaccinated annually or with some frequency, compared to 35.5% of people under 65 years of age (*p* < 0.001).

The main reasons for Spaniards to be vaccinated against influenza were their own protection and/or that of their environment and because of their doctor’s recommendation. Depending on age, the reasons “because I have enough information about the vaccine”, “because my doctor recommended it” and “because nurses recommended it” were more frequently answered by those aged 65 and over than by the rest (*p* < 0.05) (Table 2). Nonetheless, nearly 40% of the unvaccinated said that they had not been vaccinated due to a lack of recommendation or prescription. Furthermore, 30% of these respondents considered influenza not to be a serious disease (Table 3).

In total, 50.4% of Spaniards intended to be vaccinated in the 2021/2022 influenza campaign during the SARS-CoV-2 pandemic. In the case of people aged 65 years and over, the intention to be vaccinated rose to 83% compared to 42.1% of those under 65 years of age (*p* < 0.001). In addition, up to 40.7% of Spaniards who did not belong to any risk group reported that they intended to be vaccinated, compared to 81.5% of those who did belong to one (*p* < 0.001). Almost 20% of those who had not been vaccinated in the past planned to be vaccinated in the 2021/22 campaign, compared to 97.7% and 73.7% who were vaccinated annually and with some frequency, respectively (*p* < 0.001). Therefore, the COVID-19 pandemic may have had a positive influence on the intention to vaccinate against seasonal influenza 2021/22. 

In relation to employment status, 80.6% of retired people intended to be vaccinated compared to 23.6% of self-employed/entrepreneurs and 37% of students (*p* < 0.001). Those with university and secondary education also showed a lower intention to be vaccinated compared to those with only primary education (*p* < 0.001) (Table 4).

The need to protect oneself against the virus (67.4%) and to do it annually (53.8%) were the main reasons for vaccination during the 2021/22 campaign. These reasons were more frequently given by people aged 65 years and older (*p* < 0.001), while younger people cited a greater awareness of the importance of vaccination because of COVID-19 (*p* < 0.001).

In addition, one third of Spaniards who were vaccinated did so because of their confidence in vaccines and also referred to social responsibility. On the other hand, lack of recommendation (42.6%) and understanding the flu vaccine as not necessary (31.7%) were the main reasons for not getting vaccinated. In addition, the lack of recommendation reached 31.3% within the group of those aged 65 and over. Up to 23% of those who did not get vaccinated considered influenza to be a virus that was not serious. However, of those who chose not to be vaccinated, only 10.7% did not trust vaccines, and 9.1% considered the flu vaccine to be ineffective (Table 5). 

Acceptance of a specific influenza vaccine for the elderly was lower the older the respondents were (*p* < 0.001), although 73.8% of the respondents agreed that the public administration should purchase them regardless of the price. Furthermore, 50.4% of respondents considered that in the COVID-19 context, in the 2021/22 campaign, the vaccine is more important than in previous years, especially for those under 30 years of age (Table 5).

Following logistic regression analysis, age, sex, risk group membership and previous vaccination history were found to be predictors of the intention to vaccinate during the 2021/22 campaign. Those who had been vaccinated annually or frequently in the past were 29.7 times more likely to be vaccinated in the 2021/22 campaign than those who had not been vaccinated, while those in risk groups were 2.2 times more likely to be vaccinated than those without risk factors. People aged 65 years and older and women were 1.8 and 1.9 times more likely to be vaccinated than those under 65 years and men, respectively (Table 6).

### 3.3. Information and Measures to Promote Vaccination

Reminders from health personnel were the main means of information about the influenza vaccination campaign (32.9%), followed by traditional media (26.9%) and public administration (12.3%). People aged 65 years and older mainly used traditional media (53.3%) and very rarely used social media networks (2.2%) (*p* < 0.001) (Figure 1).

On the other hand, it was found that the main measures to encourage vaccination were to have information about the benefits of the vaccine and to facilitate access to Primary Healthcare Centres (PHCC) (Figure 2). In addition, measures related to getting people around them vaccinated and facilitating access to PHCC were preferred by those aged 65 years of age and older, while sending annual reminders via SMS was preferred by those under 65 years of age (*p* < 0.05).

## 4. Discussion

This study was conducted in order to assess knowledge, attitudes and practices related to influenza vaccination in the Spanish adult population before and during the COVID-19 pandemic. Thus, this paper assessed vaccine uptake and factors associated with its practice, providing interesting results that have implications for public health and regional and national vaccination policies. In addition, this study analysed how the COVID-19 pandemic may have affected the willingness to be vaccinated against influenza during the 2021/22 season.

### 4.1. Attitudes and Practices Regarding Influenza Vaccination

One third of Spaniards said in the interview that they were vaccinated against influenza annually, although 52% of people said they had never been vaccinated. Up to 34.7% of Spaniards who did not belong to any risk group said they were vaccinated annually or with some frequency, compared to 77.5% of those who did belong to a risk group. In addition, 69.1% of people aged 65 and over said they were vaccinated annually. In Europe, the influenza vaccine coverage in people over 60 years of age that were vaccinated annually varied from 50.8% in Greece [15] to 72.1% in the United Kingdom [23] in pre-pandemic times, which was slightly increased in all countries during the COVID-19 pandemic [11]. This increase during the COVID-19 pandemic is related to the pandemic, as the population is more aware of the need for vaccination to combat other viruses such as influenza. Our findings indicate that there is still room for improvement in the knowledge and health education of older people, as well as the promotion of the importance and benefits of vaccination in this age group by healthcare professionals and the media. Implementing information and education campaigns on influenza, its complications and the effectiveness and safety of vaccination, as well as designing vaccination campaigns taking into account the recommendations of health professionals, would be optimal measures to increase vaccination coverage. According to the results obtained, older age is associated with a higher probability of receiving the vaccine, which is consistent with the findings in the literature [15]. This statement may also be influenced by the health recommendations sent to the population through the media. In addition, the vaccination coverage achieved in previous campaigns together with the vaccination intention data of the present study shows that younger people are less likely to be vaccinated and have lower vaccination intentions [3]. The influence of age on influenza vaccination appears almost consistently in studies on predictors of vaccination [24,25,26]. 

Multivariate analysis showed that those in a risk group were 2.2 times more likely to be vaccinated than those with no risk factors. Since chronic illness is an indication for influenza vaccination, it is expected that vaccination is also higher in individuals belonging to risk groups [27,28]. 

The main reason respondents reported for receiving the influenza vaccine was to protect their health (71.5%), followed by their doctor’s recommendation (61%), which is consistent with what has been reported in other studies. Medical advice is significantly associated with the acceptance of vaccination [29,30,31].

Lack of recommendation and low risk perception are important barriers to vaccination. The latter reason shows the lack of information about the disease and possible complications that make vaccination necessary. Several studies indicate the possibility of improving vaccination coverage through educational initiatives targeting vaccine refusers [32,33].

The results also show that the percentage of people who intended to be vaccinated against influenza in the 2021/22 campaign during the SARS-CoV-2 pandemic (50.4%) was similar to that of those who did not intend to be vaccinated [3].

According to other studies, the intention to be vaccinated for the current season is high in people who already vaccinate annually or with some frequency [3,34], which shows that previous vaccination history is a very important predictor of vaccination in the current season (OR: 29.7; 95% CI: 22.5–39.2). If people have a positive initial vaccination experience, they are likely to seek vaccination year after year and get into the habit of being vaccinated. Among the unvaccinated, almost 20% of those who had not been vaccinated in the past planned to be vaccinated during the 2021/22 campaign, so there was a considerable increase in the percentage of people intending to be vaccinated. These data suggest that the current COVID-19 pandemic may have affected the willingness to receive influenza vaccination during the 2021/22 season among people who had not received the vaccine in the past.

A lower level of education was associated with a higher intention to be vaccinated during the 2021/22 campaign, a fact also observed in an Italian study, with higher vaccination rates in the primary education group [27].

Nearly 75% of respondents agreed to accept a specific influenza vaccine for older people in order to lessen the effects of immunosenescence on vaccine-induced immune responses in this age group.

### 4.2. Information and Measures to Promote Vaccination

Reminders from health professionals were the main means of information about the influenza vaccination campaign (32.9%), followed by traditional media (26.9%). These results are similar to those found in other studies [29], where the information came mainly from general practitioners (63.6%), followed by primary care nurses (27.8%) and the media (21.9%). In addition, receiving a reminder from a doctor could facilitate vaccination [35]. In fact, primary care physicians, who are in close contact with their patients, are best placed to provide adequate information on vaccination in accordance with existing recommendations, to answer specific questions from their patients, to send information materials and vaccination reminders and to provide information on vaccination [17]. This study found that people aged 65 and over mainly used traditional media (53.3%) and very rarely used social networks (2.2%), in contrast to younger people, so it would be advisable to adapt the communication channels used according to the different age groups.

Public administration was the third source of information used (12.3%). The role of institutions is crucial in delivering messages about vaccination, and although the information provided by institutional bodies is generally of high quality and evidence-based, web content may not be optimal in terms of comprehension.

Social media networks, used by only 9.9% of respondents to receive information about the vaccination campaign, were less trusted and could spread misinformation provided by the “anti-vaccination” movement. The high degree of vaccine questioning and reluctance to accept vaccination is amplified by social media platforms [36]. The anti-vaccination message on the internet is far more unbridled than in other media. The internet represents a more significant potential for the public to make consistent decisions about vaccination [37]. Many people search online for health information, and the information found impacts patient decision-making [38]. Given the changes in the communication landscape that fuel the rapid spread of vaccine information and misinformation, new methodologies are needed to monitor emerging vaccine concerns over time to inform appropriate responses better [36].

Accessibility to Primary Healthcare Centres is one of the main measures to encourage vaccination in the 65+ age group, because the distance and transport to these centres can be major barriers to vaccination in this age group. Therefore, improving access to vaccination by ensuring the availability of vaccines, providing dedicated vaccination sites, improving the appointment process and vaccinating in a progressive manner would be considered strategies that could favour vaccination.

There are some limitations in the design of this study. First, due to the cross-sectional design, it is not possible to assign causality between vaccination and related variables. In addition, the data collected in the questionnaire were obtained directly from the participant’s responses, so the results may be affected by recall bias or misinterpretation of the questions by the respondents. On the other hand, a strength of this study is that it was a representative sample of the population as the data were obtained from a nationwide survey. Moreover, we believe that this study can help to improve policy decisions in the area of health recommendations by strengthening the promotion of vaccines among the population.

## 5. Conclusions

The benefits of influenza vaccination are widely evidenced, and it is the most effective way to prevent the disease and its complications, especially among the most vulnerable groups. In addition, this work demonstrated that the COVID-19 pandemic in Spain has likely had some impact on the national level of influenza vaccination coverage during the 2021/22 season. This study aimed to assess and analyse the factors influencing willingness to receive influenza vaccines in the COVID-19 era among Spanish adults, enabling an analysis of the potential points to be conducted to further improve vaccination coverage, which would translate into health benefits for those vaccinated and their environment, as well as economic benefits for the system and society. The recommendations of health professionals, as well as greater education and awareness of influenza, are key aspects to encourage vaccination. It is important to insist on the need to vaccinate the elderly, as the lack of recommendation reached 31.3% in the group of those aged 65 and over. It is also necessary to continue research into the measures that could be most effective, as well as analysing the effect of the pandemic in the medium and long terms on influenza vaccination. 

## Figures and Tables

**Figure 1 ijerph-19-11139-f001:**
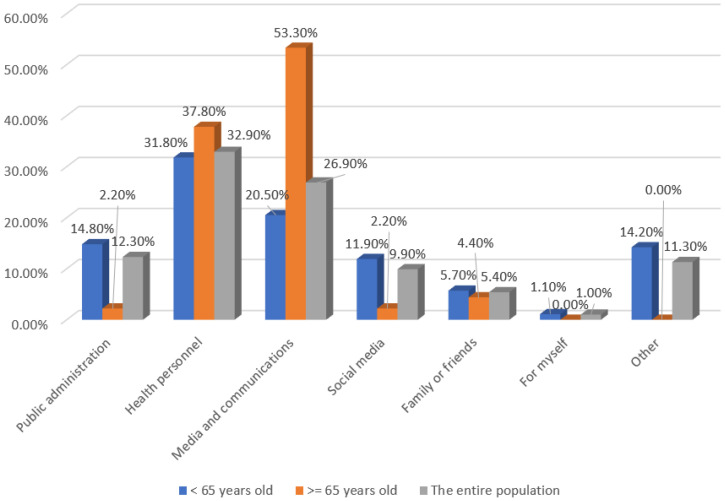
Main means of information about the influenza vaccination in the adult population and by age.

**Figure 2 ijerph-19-11139-f002:**
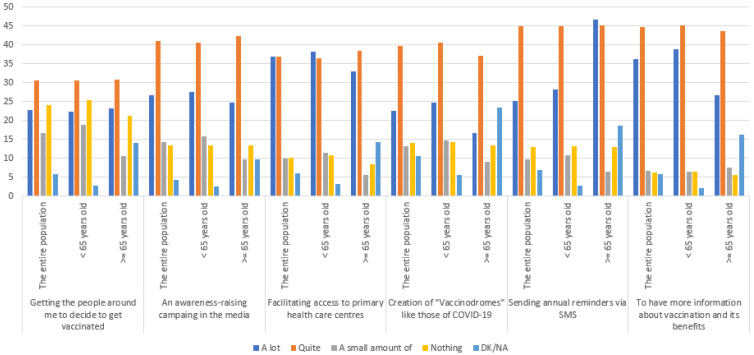
Measures to promote vaccination.

**Table 1 ijerph-19-11139-t001:** Socio-demographic characteristics of the study participants.

Variables	Values	N (%)
Gender	Women	1139 (51.7)
	Men	1066 (48.3)
Age group	18–29	299 (13.6)
	30–44	507 (23.0)
	45–64	803 (36.4)
	65 or more	596 (27.0)
Employment status	Private sector worker	659 (29.9)
	Public sector worker	242 (11.0)
	Self-employed/entrepreneur	131 (5.9)
	Retired or pensioner	669 (30.4)
	Unemployed	268 (12.1)
	Student	94 (4.3)
	Unpaid domestic work	142 (6.4)
Level of education	Primary or lower	710 (32.2)
	Secondary	869 (39.4)
	University	626 (28.4)
Membership of risk groups	Over 65 years of age	249 (37.0)
	Respiratory pathologies	115 (17.1)
	Diabetics	79 (11.8)
	Cardiovascular diseases	66 (9.8)
	Immunosuppression	19 (2.9)
	Other chronic pathologies	87 (12.9)
	Health professionals	27 (4.0)
	Essential workers ^1^	17 (2.5)
	Other ^2^	14 (2.1)

^1^ Essential workers are considered to be those indicated by the government Royal Decree-Law 10/2020 of 29 March published by the Government of Spain [22]. ^2^ Within the category “other”, there are three risk groups: people with disabilities, pregnant women and people living with at-risk patients.

**Table 2 ijerph-19-11139-t002:** Attitudes and practices in relation to previous flu vaccination level.

Variable	Values	Annual or Frequent Vaccinations	N/Total (%)
			*p*
Risk group	Yes	521/672 (77.5)	<0.001
	No	525/1511 (34.7)	
Age	≥65	485/594 (81.6)	<0.001
	<65	570/1605 (35.5)	

**Table 3 ijerph-19-11139-t003:** Reasons why people chose to get vaccinated against flu or not in previous campaigns ^1^.

	Reasons	N (%)
Reasons for deciding to vaccinate (N = 1052)	For my own protection and/or environment	755 (71.5)
	Because my doctor recommended it	642 (61.0)
	Because of social responsibility	296 (28.1)
	Because I have suffered from the consequences of influenza in the past	180 (17.1)
	Because I have sufficient information about the vaccine	176 (16.7)
	Because it was recommended to me by nurses	130 (12.3)
Reasons for deciding not to vaccinate (N = 1132)	Not recommended/prescribed	437 (38.6)
	My doctor has not recommended it	411 (36.3)
	I do not consider the flu to be serious	345 (30.5)
	I do not have enough information about the vaccine	115 (10.2)
	Lack of confidence in the effectiveness of the vaccine	93 (8.2)
	I have had a previous vaccination and it made me feel sick	78 (6.9)
	I have a phobia of needles	42 (3.7)

^1^ Respondents who had been vaccinated in previous campaigns (N = 1052) were allowed to choose several reasons as to why they did so. Similarly, those who did not vaccinate in previous campaigns (N = 1132) were allowed to select several reasons as to why they did not vaccinate.

**Table 4 ijerph-19-11139-t004:** Attitudes and practices regarding intention to get vaccinated against flu in the 2021/22 campaign during the COVID-19 pandemic.

Variable	Values	Intention to Vaccinate in the 2021/22 Campaign	
		N/Total (%)	*p*
Risk group	Yes	517/634 (81.5)	<0.001
	No	586/1439 (40.7)	
Age	≥65	473/570 (83)	<0.001
	<65	639/1517 (42.1)	
Vaccination in recent years	Yes, annually	682/698 (97.7)	<0.001
	Yes, with some frequency	224/304 (73.7)	
	No	204/1081 (18.9)	
Employment status	Private sector worker	255/626 (40.7)	<0.001
	Public sector worker	103/222 (46.4)	
	Self-employed/entrepreneur	30/127 (23.6)	
	Retired or pensioner	511/634 (80.6)	
	Unemployed	103/249 (41.4)	
	Student	34/92 (37.0)	
	Unpaid domestic work	76/138 (55.1)	
Level of studies	Primary or lower	441/679 (64.9)	<0.001
	Secondary	373/810 (46.0)	
	University	298/598 (49.8)	

**Table 5 ijerph-19-11139-t005:** Reasons why people expressed that they were hesitant to get vaccinated against flu or not in the 2021/2022 campaign ^1^.

	Reasons	N (%)
Reasons for deciding to vaccinate (N = 1109)	You need to protect yourself against viruses	748 (67.4)
	It is necessary to get an annual flu vaccination	596 (53.8)
	I trust vaccines in general	403 (36.7)
	Because of social responsibility	358 (32.5)
	Because COVID has made me more aware of the importance of vaccination.	239 (21.7)
	In case I have COVID, it will help me with the effects of the vaccine.	212 (20.0)
Reasons for deciding not to vaccinate (N = 1070)	Not recommended/prescribed	456 (42.6)
	No vaccination is necessary	335 (31.7)
	I do not consider flu to be a serious (deadly) virus	246 (23.2)
	I do not trust vaccines in general	116 (10.9)
	COVID-19 vaccine is sufficient	112 (10.7)
	Not effective	96 (9.1)

^1^ Respondents who answered that they intended to be vaccinated in the 2021/2022 campaign (N = 1109) were allowed to select several reasons as to why they would do so. Similarly, those who did not intend to vaccinate in the 2021/2022 campaign (N = 1070) were allowed to select several reasons as to why they would not.

**Table 6 ijerph-19-11139-t006:** Predictors of vaccination intention in the 2021/22 campaign.

Variables	Categories	*p*	OR	95% CI
Gender	Woman Man	<0.001	1.9 1	1.5–2.4
Age	≥65 <65	0.001	1.8 1	1.3–2.5
Risk group membership	Yes No	<0.001	2.2 1	1.6–3.1
Previous vaccination	Yes No	<0.001	29.7 1	22.5–39.2

## Data Availability

The data presented in this study are available on request from the corresponding author. The data are not publicly available due to privacy policies of the company that carried out the surveys.

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
