# Peer review of "Attitudes, Perceptions and Practices of Influenza Vaccination in the Adult Population: Results of a Cross-Sectional Survey in Spain"

_ijerph, 2022, doi:10.3390/ijerph191711139_

Round 1
Reviewer 1 Report
I consider the research work to be pertinent as it responds to a relevant research question such as influenza vaccination coverage in the adult population, one of the most important public health strategies that is carried out every year and which always has important areas for improvement as we do not reach the coverage established by international organisations such as the World Health Organisation.
The results are well presented and well discussed, although it would be interesting to incorporate some bibliographical citations in the discussion that deal with this same subject in your geographical environment, so I recommend that you incorporate some citations like the ones I provide or look for other similar and recent ones.
I consider that the work can be published and in any case expand the discussion a little with some bibliographic citations like the ones I am sending you:
Díez-Domingo J, Redondo Margüello E, Ortiz de Lejarazu Leonardo R, Gil de Miguel Á, Guillén Ortega JM, Rincón Mora J, Martinón-Torres F. A tool for early estimation of influenza vaccination coverage in Spanish general population and healthcare workers in the 2018-19 season: the Gripómetro. BMC Public Health. 2022 Apr 25;22(1):825. doi: 10.1186/s12889-022-13193-x. PMID: 35468772; PMCID: PMC9036844.
Reviewer 2 Report
Review comments
The manuscript presents information from a national survey that was conducted in 2021 in Spain to understand perceptions, attitudes and planned practices related to influenza vaccine in the general population. The method used for the study is sound and the main conclusions are relevant and worth publishing. However, there are problems with the current manuscript should be addressed.
Minor comments
1. Abstract needs revision structuring its content on objectives, methods, main results and conclusions and avoid redundancy in the presentation of results or the aims of the study. The results related to one of the main objectives of the study: health care providers as source of information should also appear on the abstract.
2. Table 1 label is repeated (it is wrongly label as Table 1 what appears to be Table 3). in the text,
3. Figures 1 and 2 should be combined and present simultaneously separate values for the overall population and by age group.
4. Figure 3 should also present information separately by age group
Major issues
1. The main document is well structured and provides interesting and relevant information. However, the text requires English editing in order to make it fluid and amenable to read.
2. Wording in the text should be reviewed to make it more simple and direct, and relevant to the aims and hypothesis of the study. For example categories for employment status are too specific making it unnecessarily ambiguous and confusing. For example: “… in retirement or retired”, “cardiac pathologies” probably meaning cardiovascular diseases, and so on. “Essential workers” should be explained as footnote to the table.
3. Tables 1, 2 and 3 should be redesign to provide more information. Heading and wording of columns, rows, and so on has to be revised as well, as stated earlier. Some categories on Table 1 are too small and could be combined on “Other” and detail composition in a footnote in the table.
Tables 2 and 3 should include the actual number of people in each category of analysis. For example, how many people received annual or frequent vaccination in each group and how many did not. Providing the p value is not sufficient, as it does not allow readers to understand the real differences or reproduce the analyses.
Reviewer 3 Report
Many thanks for the opportunity of reviewing this important work by Prada-García and colleagues, which aims to determine the knowledge The manuscript is well-written and clearly presented.
I have some suggestions before the manuscript can be published:
- the references used in the introduction do not seem appropriate as they do not match the contents (e.g. "Globally, annual epidemics are estimated to cause 3-5 million cases of severe disease and 250-500k deaths per year [2,7-10]"). In 2,7,8,9,10 you mainly report studies that investigated the role of some factors in predicting influenza vaccination. I suggest revising the reference list; in this specific case, I suggest mentioning studies that really investigated (globally or regionally) the burden of disease brought by influenza infections [examples of references are given below]
- In the discussion, (line 262-264) you correctly report influenza vaccination coverage rates in Europe, but only mention coverage rates in 2020, before the COVID-19 pandemic. I suggest mentioning what changed during the last season, briefly discussing the reasons why this may have happened [examples of references are given below]
- I suggest stating clearly that some findings may be biased/influenced by the current recommendations. One example: since influenza vaccination is recommended everywhere to the elderly, I think that there is no point in saying that "older age is associated with a higher probability of receiving the vaccine", unless you mean that also in the same age group (e.g. over 65) those who are older (e.g. over85) get vaccinated more than the others (e.g. under85).
- I am missing a final point of the discussion section in which you explain how a policy maker could use important findings (e.g. to promote vaccinations or re-organize vaccinations, or change the policies). I think a reader would benefit from such a paragraph.
Good luck!
Paget J, Spreeuwenberg P, Charu V, et al. Global mortality associated with seasonal influenza epidemics: New burden estimates and predictors from the GLaMOR Project. J Glob Health. 2019;9(2):020421. doi:10.7189/jogh.09.020421
Iuliano AD, Roguski KM, Chang HH, et al. Estimates of global seasonal influenza-associated respiratory mortality: a modelling study [published correction appears in Lancet. 2018 Jan 19;:]. Lancet. 2018;391(10127):1285-1300. doi:10.1016/S0140-6736(17)33293-2
Del Riccio M, Lina B, Caini S, et al. Letter to the editor: Increase of influenza vaccination coverage rates during the COVID-19 pandemic and implications for the upcoming influenza season in northern hemisphere countries and Australia. Euro Surveill. 2021;26(50):2101143. doi:10.2807/1560-7917.ES.2021.26.50.2101143
